# The Microbiome of Things: Appliances, Machines, and Devices Hosting Artificial Niche-Adapted Microbial Communities

**DOI:** 10.3390/microorganisms11061507

**Published:** 2023-06-06

**Authors:** Leila Satari, Alba Iglesias, Manuel Porcar

**Affiliations:** 1Institute for Integrative Systems Biology (I2SysBio), Universitat de València-CSIC, 46980 Paterna, Spain; 2Darwin Bioprospecting Excellence SL., Parc Científic, Universitat de València, 46980 Paterna, Spain

**Keywords:** artificial environments, man-made devices, selective pressure, microbiome of things (MoT), adaptive mechanisms

## Abstract

As it is the case with natural substrates, artificial surfaces of man-made devices are home to a myriad of microbial species. Artificial products are not necessarily characterized by human-associated microbiomes; instead, they can present original microbial populations shaped by specific environmental—often extreme—selection pressures. This review provides a detailed insight into the microbial ecology of a range of artificial devices, machines, and appliances, which we argue are specific microbial niches that do not necessarily fit in the “build environment” microbiome definition. Instead, we propose here the Microbiome of Things (MoT) concept analogous to the Internet of Things (IoT) because we believe it may be useful to shed light on human-made, but not necessarily human-related, unexplored microbial niches.

## 1. Microorganisms: From General to Specialized Metabolisms

Earth is home to a large biodiversity of microorganisms, recently predicted to be as high as 10^12^ or more microbial taxa [1]. For decades, natural environments have been explored to study the microbial communities that inhabit them [2], as well as the specific adaptation mechanisms of the microbial strains living there and their biotechnological potential. A clear example is the study of soil microorganisms capable of producing molecules with plant growth-promoting or antimicrobial activities [3,4]. Not only natural but also artificial environments constantly manipulated by human activities hold interest as sources of microbial diversity. These environments include solar salterns, wastewaters, and, more recently, nuclear waste repositories [5,6,7].

Artificial products, such as machines, home appliances, electronic devices, etc., are generally very persistent in the environment [8]. The microbial populations that thrive in these kinds of waste have received more industrial attention during the last few years, as several studies have shown that specific microbial populations can colonize the surface of those products [9,10] and, in some cases, gradually modify and decompose those materials [10,11]. An interesting example is the wasted chewing gum, composed mainly of synthetic polymers and elastomers, in which the initial microbiome, dominated by bacteria from the human mouth cavity, is substituted after a few weeks of outdoor exposure by environmental bacteria that may participate in the long-term biodegradation of the wasted chewing gum [12]. Another example is the biodegradation of plastics, in which many environmental microorganisms have been reported to exhibit a modest to moderate biodegradation ability on various types of plastics, such as polyethylene (PE), polypropylene (PP), polyethylene terephthalate (PET), polypropiolactone (PPL), polycaprolactone (PCL), Polybutylene succinate (PBS), and polyvinyl chloride (PVC) [9,13,14]. The aim of this review is to shed some light on the microbiomes of a variety of artificial devices, focusing on how the specific pressure of a given artificial niche shapes the microbial community inhabiting it, a perspective that has allowed us to propose the concept of Microbiome of Things (MoT).

## 2. Microbiomes on Artificial Devices

Machines and environments associated with humans, such as elevators [15], underground subways [16,17], and small electronic devices [18,19], are rich microbial niches. This review, however, will focus on artificial appliances, machines, and devices with their own associated microbiome, rather than large structures or “indoor” environments. Here, we describe the microbial profile of “things”, which, despite being in constant contact with humans or the human activity, do not necessarily share a microbiome characterized by human-associated microorganisms but, instead, represent microbial micro-niches with their own selective pressures and characteristic microbiomes. Here, we look briefly into the previous research on the natural microbiomes of “things”, and we use examples such as that of the photovoltaic panels [20,21], the car tank lid [22], or the microbiome of automobile air-conditioning systems [23]. The focus of the “Microbiome of Things” (MoT), the existence of which we propose here, is analogous to the “Internet of Things” (IoT) concept. Our definition of “thing”, though, is not necessarily an internet-connected machine, as is the case for IoT, but rather an *ad hoc* one: we consider a “thing” any artificial machine, device, or artifact that is typically a few centimeters to a few meters in length and that is susceptible to significant microbial colonization. Essentially, all “things” are artificial devices in constant contact with humans or the human activity, but not all artificial devices are considered “things”. Only those devices or machines whose characteristics result in a selective pressure display a MoT and, as the reader will see in this review, these uniquely adapted MoT constitute a rich source of biomolecules with biotechnological potential.

### 2.1. Microbial Diversity of Sun-Exposed Artificial Outdoor Surfaces

**Photovoltaic panels** are smooth, glass, or glass-like sun-exposed surfaces with a minimum water retention capacity. Microorganisms inhabiting them have developed specific adaptation mechanisms, such as, for example, the production of adhesins for attachment or antioxidants to survive irradiation and desiccation. Although not in close contact with humans once in place, previous works have made it clear that solar panels are not merely accumulating dust-borne microorganisms, but, instead, a selection takes place in situ, demonstrating that it is their own nature, rather than the climate, that shapes the microbial community inhabiting them [20,24]. Interestingly, regardless of the geographical location and climate differences in different locations of the world, there is a shared microbial community inhabiting the surface of solar panels, dominated by *Hymenobacter* spp., *Sphingomonas* spp. and *Deinococcus* spp. [20,24,25], and, to a lesser extent, *Methylobacterium* spp. [21], *Roseomonas* spp. [26] and *Novosphingobium* spp. [25]. Members from the genus *Sphingomonas* are among the most abundant strains inhabiting sun-exposed surfaces such as solar panels [27] and have a crucial role in the early steps of microbial colonization and biofilm formation [28]. In addition, the presence of sphingolipids, such as glycosphingolipids in some species (*Sphingomonas* spp.), has been described as a facilitator of the bacterial attachment to polyamide and silica [29]. The photovoltaic panel microbiome is very likely shaped by the strong selection pressure of UV irradiation; besides, other factors, such as the limitation of nutrients, desiccation, and dramatic temperature changes, may also play a major role in selecting for specific resident microbial taxa [26]. To date, several studies have shown that most culturable bacteria isolated from photovoltaic panels can produce a wide range of carotenoids to tolerate radiation [20,26]. Indeed, the antioxidant-related metabolisms of members of the genera *Hymenobacter* and *Methylobacterium* suggests that these microorganisms can have promising applications in the food and pharmaceutical industries [21].

### 2.2. Microbial Diversity of Indoor Artificial Devices and Appliances

Artificial electronic devices, such as the surface of smartphones or elevator buttons, are inhabited by both the human microbiome and environmental microorganisms [15,30]. Briefly, although the human microbiome is highly personalized and the skin microbiome varies between sites and skin types [31], bacteria that thrive in the skin of healthy adults include, but are not limited to, *Propionibacterium*, *Staphylococcus*, and *Corynebacterium* species. Although less abundant than bacteria, fungi of the genera *Malassezia*, *Aspergillus*, *Cryptococcus*, *Rhodotorula*, and *Epicoccum* appear across all surface body sites [32]. Even though most built environments have received attention for being colonized by human-associated microorganisms of public health concern, most of these communities are dominated by environmental microorganisms [33]. In the case of the built environment, members of *Pseudomonadaceae*, such as *Acinetobacter* and plant-associated bacteria such as *Neorhizobium*, were found to predominate in kitchens, whereas skin-associated bacteria—for example, *Staphylococcus*, *Propionibacterium*, *Corynebacterium*, and *Streptococcus*—dominated in the bathrooms [34].

Although, ideally, to truly distinguish MoT from the human and built-environment microbiomes, it would be necessary to include the kind of studies we have analyzed in this review and negative controls from the outer surfaces of the devices or nearby objects. Previous work has also demonstrated that the microbial exchange between the human skin and a built environment can result in the formation of an independent microbial community after some time [35]. The analysis of the bibliography clearly suggests that artificial devices with a selective pressure tend to have additional non-human-associated, specifically adapted microbial taxa that result in truly unique microbial niches. The most common microbial taxa that colonize the following artificial devices are summarized in Table 1, although it should be taken into account that for some artificial niches there were no studies of the fungal communities.

**Coffee machines:** A one-year study of the microbiome of the wasted capsule dip tray of ten coffee machines running on capsules and based on the 16S rRNA gene amplicon analysis showed that their microbial profiles were relatively similar in all the analyzed coffee machines [36]. *Enterococcus* and *Pseudomonas* were the most frequent shared taxa, followed by other abundant genera such as *Stenotrophomonas*, *Sphingobacterium*, *Acinetobacter*, *Coprococcus*, *Paenibacillus*, or *Agrobacterium*. The colonization process of the wasted coffee leach was monitored over two months [36]. Microbiome sequencing of the 16S rRNA gene revealed that, during the first two weeks, *Pantoea* sp., *Cloacomonas* sp., and *Brevundimonas* sp. were the most frequent taxa, whereas those taxa were substituted by species from the genera *Pseudomonas*, *Acinetobacter*, and *Sphingobium* later. Species of the genera *Pseudomonas* and *Enterococcus* dominated the bacterial community after two months. These results suggested that abiotic conditions such as temperature and nutrient availability changes, as well as the accumulation of caffeine, may be the key factors shaping the final caffeine-adapted microbial community [36].

**Dishwashers:** Recent research has studied the microbial colonization of dishwashers, another man-manufactured device subjected to extreme selective pressures [37,38]. Previous research revealed that dishwashers can be considered poly-extreme habitats for microorganisms due to thermal stress, high salt concentrations, presence of detergents, pH variations, and constant water pressure [39]. The microbial communities that colonize these devices form biofilms not only on the surface of rubber seals but also on entire interior surfaces [40]. Although, for sanitary reasons, most efforts have focused on monitoring the presence of opportunistic microorganisms in dishwashers [37,41,42,43], a small but important fraction of the dishwasher microbiome consists of environmental microorganisms such as *Gordonia* spp., *Micrococcus* spp., *Chryseobacterium* spp., *Exiguobacterium* spp., and *Meiothermus* spp. [39]. These microorganisms were previously reported as halotolerant, heavy metal tolerant, UV-resistant, and thermotolerant microorganisms [44,45,46,47]. Moreover, besides opportunistic microbes, some human-associated and opportunistic bacteria, such as *Staphylococcus* sp., *Streptococcus* sp., *Lactobacillus* sp., *Corynebacterium* sp., *Enterococcus* sp., *Acinetobacter* sp., *Escherichia*/*Shigella* sp., and *Pseudomonas* sp., have been commonly detected in the dishwasher biofilms [39,40]. Although fungi from the phylum *Ascomycota*, followed by *Basidiomycota*, tend to be the first colonizers of the rubber surfaces, previous research showed that the fungal communities can be affected by the high pressure of the water pumping system [40]. For the formation of biofilms, one essential factor is the microbial extracellular polymeric substances (EPSs) responsible for assisting individual cell immobilization, accelerating cell-to-cell interactions, and, finally, synergizing the formation of the microbial consortia [48]. Dishwasher microorganisms such as *Gordonia* sp., *Micrococcus* sp., *Chryseobacterium* sp., *Exiguobacterium* sp., and *Acinetobacter* sp. have previously been reported as EPSs producers [47,49,50,51,52].

**Washing machines:** Biofilm formation and microbial survival on the interior and exterior surfaces of washing machines have been studied [53,54]. Similar to the extreme selective conditions of the dishwashers, the microbial communities within washing machines are able to tolerate a diverse range of temperatures, pHs, and detergent concentrations. Previous research revealed that the microbial community of a washing machine significantly depends on the site of sampling and the number of high-temperature wash cycles per month [55]. For example, the composition of the microbial communities in the detergent drawer is different to that of the tub, due to the enzymatic effect that concentrated detergents have on some microorganisms. Species from the genera *Pseudomonas*, *Acinetobacter*, and *Enhydrobacter* dominate the washing machine microbial communities regardless of the sampling site [55]. The same study revealed that human pathogens were present at low frequency in the washing machine microbiome. Another work by Callewaert *et al*. (2015) corroborated that the presence of some pathogens, such as *Leptospira* or *Legionella* sp., is less than 1% of the total microbial community [56]. The presence of members from the *Moraxellaceae* family was also detected, and previous studies demonstrated that bacteria such as *Moraxella* sp. cause the characteristic malodor of clothes after a suboptimal laundry cycle [57,58].

**Water heating systems:** In the early 1970s, Brock and Boylen revealed that some environmental extremophiles can thrive in the extreme conditions of water heating (WH) systems [59]. Although, similar to solar panels, WHs could have limited exposure to human activity or the human microbiome and harbor instead environmental microorganisms characteristic of water microbiomes, this is another clear example of how the pressures of these devices play a key role in selecting their own microbiome. *Thermus aquaticus* was isolated from hot-water tanks after being found in natural hot springs. Interestingly, the *T. aquaticus* strains isolated from hot-water heaters did not show the pigmentation of the strains isolated from natural environments [59], which may be linked to the role of pigments for quenching of singlet oxygen in intense sun-exposed irradiated environments [60]. Later, Kjellerup *et al*. (2005) studied biofilm formation in WHs and demonstrated that the parameters playing a key role in the development of microbial biofilms were pH, conductivity, and oxygen concentration [61]. The result of FISH analysis revealed that ß-proteobacteria was the most abundant bacteria, followed by sulfate-reducing bacteria (SRB), ɑ-proteobacteria, and, to a lesser extent, ɣ-proteobacteria. At the genus level, the presence of *Acidovorax* sp., *Agrobacterium* sp., *Roseococcus* sp., *Flavobacterium* sp., and, to a lesser extent, *Sphingomonas* sp., were monitored in the heating system biofilm by DGGE analysis [61].

**Saunas** reach temperatures of approximately 75 to 80 °C. Typical human-associated bacteria cannot thrive in such a hot environment. However, some spore-forming microorganisms transiently present in the skin or other surfaces can tolerate harsh and unfavorable conditions and can survive very high temperatures, even in the presence of detergent and disinfectants [62]. Previous research by Lee *et al*. (2012) showed that spore-forming bacteria such as *Bacillus* sp., *Virgibacillus* sp., and non-spore-forming thermophilic bacteria like *Tepidomonas* sp., and *Pseudoxanthomonas* sp., as well as non-thermophilic bacteria belonging to the genera *Stenotrophomonas* and *Janthinobacterium*, were detected by TGGE [62]. In other research, Kim *et al*. (2013) studied the bacterial diversity of two dry saunas that were operated at lower and higher temperatures (64 °C and 76 °C, respectively). Members of the genera *Moraxella* and *Acinetobacter* were detected in the low-temperature sauna, whereas strains belonging to the genera *Aquaspirillum*, *Chromobacterium*, *Aquabacterium*, *Gulbenkiania*, *Pelomonas*, and *Aquitalea* dominated the high-temperature sauna. Thermophilic strains such as *Bacillus megaterium* and *Deinococcus geothermalis* were also found in both low- and high-temperature saunas [63]. In another report, Tanner *et al*. (2017) reported a similar taxonomic distribution in another sauna sample characterized by a high abundance of ɑ-, ß-, and ɣ-proteobacteria, and, to a lesser extent, *Bacteroidetes*, *Actinobacteria*, and *Acidobacteria*. Scanning Electron Microscopy (SEM) of the sample showed a dense microbial biofilm within a smooth matrix of EPS. By culture-dependent techniques, a moderate number of bacteria with thermophilic lipolytic activity were isolated in that study [64].

**Refrigerators:** In refrigerators, moisture and nutrients from food particles provide a favorable ecosystem for microorganisms to thrive. Although in constant contact with food and the human skin microbiota, the low temperature (4 °C) also limits microbial growth. *Proteobacteria*, *Firmicutes*, *Actinobacteria*, and *Bacteroidetes* were predominant in refrigerator samples, and *Pseudomonas* and *Pantoea* were the dominant genera [65]. Moreover, this study revealed that, on average, the bacterial communities in refrigerators shared 15.6% of the bacterial species on human skin and 4.9% of the species from the human gut. Another more recent study analyzed the microbial population in the cold storage room of domestic refrigerators. In this case, members of the families *Enterobacteriaceae* and *Bacillaceae* predominated in all the samples, *Bacillus* and *Acinetobacter* being the most abundant genera, although *Enterococcus*, *Citrobacter*, *Exiguobacterium*, *Staphylococcus*, *Enterobacter*, and *Pseudomonas* were also present in the refrigerators analyzed [66]. This study also revealed the fungal community of refrigerators, which was dominated at the genus level by *Saccharomyces* and *Candida* in 75% of the refrigerators under study, although members of the genera *Malassezia*, *Schwanniomyces*, and *Kazachstania* were also detected [66].

**Air conditioning systems (ACs)** are another type of indoor device that harbors microbial communities [67,68]. In ACs, warm and humid air passes through the cooling coils as it cools down, leading to water condensing on the coil surfaces [69]. Research by Acerbi *et al*. (2017) showed that the microbial populations from the air and the condensed coil water were significantly different, and *Agaricomycetes* was the most abundant taxa in both. The results showed that the air samples were more diverse than the condensed water samples. The *Moraxellaceae* family (including genera such as *Perlucidibaca* or *Acinetobacter*) and *Enhydrobacter*, followed by *Pseudomonas* spp., were also found in the air samples. In the same study, the population of *Sphingomonas* spp. proved to increase over time and dominate the microbiome at the late phase of the study [69]. Another report on the microbial diversity of forty large-scale commercial ACs showed that *Methylobacteriaceae* and *Propionibacterium* spp., and, to a lesser extent, *Acetobacteraceae* and *Sphingomonas* spp., dominated the ACs’ bacteriome [70]. The analysis of the fungal communities within those forty commercial ACs showed no specific pattern to the distribution of fungal taxa, although the presence of *Malassezia* spp., *Cladosporium* spp., and *Leotiomycetes* was confirmed, with different relative abundances [70]. Interestingly, previous works also demonstrated that *Methylobacterium* sp. and *Propionibacterium* sp. can grow in oligotrophic environments such as the surface of the cooling coils in air-handling systems and air conditioning systems of automobiles by biofilm formation [71,72]. The presence of *Malassezia* spp. is not surprising, either, as this common skin-commensal yeast has been widely reported in indoor environments [73]. Another study by Wilson *et al*. (2007) reported that spore-forming *Cladosporium* sp. dominated the microbial communities of various parts of air-handling systems [74].

**Table 1 microorganisms-11-01507-t001:** Most abundant microorganisms colonize man-made devices under different selective pressures such as thermal stress, pH variations, desiccation, UV-irradiation, *etc*.

Artificial Devices	Extreme Condition of the Device	Most Abundant Genera	Ref.
**Photovoltaic panels**	UV-irradiation Desiccation Low nutrition	**Bacteria:** *Hymenobacter* *Sphingomonas* *Deinococcus* *Methylobacterium* *Roseomonas* *Novosphingobium*	[20,21,24,25,26]
**Coffee machines**	Thermal stress Water pressure Particular nutrient availability Alkaloids (caffeine)	**Bacteria:** *Enterococcus* *Pseudomonas* *Stenotrophomonas* *Sphingobacterium* *Acinetobacter* *Coprococcus* *Paenibacillus* *Agrobacterium* *Sphingobium*	[36]
**Dishwashers**	Thermal stress High salt concentrations Presence of detergents pH variations Water pressure	**Fungi:** *Ascomycota* *Basidiomycota* **Bacteria:** *Gordonia* *Micrococcus* *Chryseobacterium* *Exiguobacterium* *Meiothermus* *Staphylococcus* *Streptococcus* *Lactobacillus* *Corynebacterium* *Enterococcus* *Acinetobacter* *Escherichia*/*Shigella* *Pseudomonas*	[39,40]
**Washing machines**	Thermal stress Presence of detergents pH variations Constant water pressure	**Bacteria:** *Pseudomonas* *Enhydrobacter* *Leptospira* *Sphingomonas* *Legionella* *Moraxellaceae* family *Acinetobacter*	[55,56,57,58]
**Water heating systems**	High temperature Conductivity and oxygen concentration	**Bacteria:** *Thermus* *Acidovorax* *Agrobacterium* *Roseococcus* *Flavobacterium* *Sphingomonas* *Brochothrix* *Buchnera* *Polynucleobacter* *Ralstonia* *Thermicanus* *Parascardovia* *Micrococcus* *Rothia* *Brachybacterium* *Methylobacterium* *Sejonia* *Moraxellaceae* family	[59,61,75]
**Saunas**	High temperature Low nutrient	**Bacteria:** *Bacillus* *Virgibacillus* *Tepidomonas* *Pseudoxanthomonas* *Stenotrophomonas* *Janthinobacterium* *Aquaspirillum* *Chromobacterium* *Aquabacterium* *Gulbenkiania* *Pelomonas* *Aquitalea* *Deinococcus* *Moraxellaceae* family	[62,63,64]
**Refrigerators**	Low temperature	**Fungi:** *Saccharomyces* *Candida* **Bacteria:** *Pseudomonas* *Pantoea* *Bacillus* *Acinetobacter* *Enterococcus* *Citrobacter* *Exiguobacterium* *Staphylococcus* *Enterobacter*	[65,66]
**Air conditioning systems**	Low temperature Low nutrient	**Fungi:** *Malassezia* *Cladosporium* *Leotiomycetes* **Bacteria:** *Agaricomycetes* *Pseudomonas* *Sphingomonas* *Propionibacterium* *Methylobacterium* *Enhydrobacter* *Moraxellaceae* family *Perlucidibaca* *Acinetobacter*	[69,70]

## 3. How to Live in a Machine: Microbial Adaptations

Among all the common survival mechanisms, biofilm formation, sporogenesis, and pigment production play a key role in the colonization of the artificial surfaces described before.

**Biofilm formation** is one of the most predominant microbial strategies to tackle multi-stress conditions in natural and artificial environments. Biofilm improves stress tolerance and biomass production and increases signaling and metabolic cooperation within a heterogenic community (Figure 1). In a multi-layer biofilm, microbial interactions affect the biochemical resources and availability within the community [76]. Biofilm formation also acts as a protective mechanism for the survival and reproduction of its members, especially under oligotrophic conditions [77]. Biofilm formation also provides efficient nutrient recycling through the facilitation of syntropy and metabolite exchange between the microbial partners. Moreover, among other parameters, biofilm formation and composition are highly influenced by the surface material. For example, metal surfaces are subjected to biocorrosion by sulfate-reducing bacteria, sulfur-, iron-, and manganese-oxidizing bacteria, and other microorganisms secreting organic acids, such as oxalate. In the case of glass, bacteria and fungi can thrive through the redox transformations of Fe, S, and Mn, and for wood- or plastic- derived materials, the extracellular enzymatic attack results in the microbial uptake of the degradation products for cell growth and development [78].

**Sporogenesis** is another strategy that allow microorganisms to survive as vegetative dormant cells for very long periods of time [79]. The dormant cells can tolerate multi-stresses such as extreme temperatures, desiccation, nutrient starvation, organic chemicals, hydrolytic enzymes, oxidizing, and DNA-damaging agents such as UV and gamma radiation [80]. Sporogenesis has been described in a broad range of *Firmicutes*, recently renamed as *Bacillota*, especially those belonging to the genus *Bacillus*. As described before in this review, different genera from *Bacillota* inhabit the high-temperature saunas and sun-exposed solar panels. Sporogenesis can thus help *Bacillus* and other sporulated microorganisms survive extreme temperatures and nutrient starvation, typical conditions of those environments.

**Colony pigmentation:** Carotenoids are the most important natural pigments produced by microorganisms [81]. They are mostly biosynthesized through the isoprenoid biosynthetic pathways, mainly recognized as C40 lipophilic isoprenoids, produced by some microorganisms, such as algae, yeast, fungi, bacteria, and haloarchaea [82,83]. Carotenoids are colorless to yellow, orange, and red pigments and are considered antioxidant compounds that play a crucial role in protecting cells, especially under intense irradiation, by quenching of singlet oxygen [60,84]. However, the role of carotenoids is not only limited to saving microorganisms from photodynamic death; they can also protect the cells by reducing hydroperoxides into stable compounds, preventing free radicals’ formation, and inhibiting the auto-oxidation chain reaction. Moreover, carotenoids can act as metal chelators and convert iron and copper toxic ions into safe molecules [85]. Furthermore, other research has shown that pigment production in multi-stress environments might be coupled to the release of some osmoregulation molecules, helping microorganisms better cope with salinity and desiccation and prevent cells from harmful damages [86]. Pigment biosynthesis has been previously reported in various bacteria, including species from the genera *Deinococcus*, *Sphingomonas*, *Methylobacterium*, *Acinetobacter*, and *Micrococcus*, among the marker microorganisms that can colonize machines, suggesting that pigment production is one of the microbial mechanisms to survive in man-made devices.

**Thermal and cold resistance strategies,** including expression of particular proteins [e.g., cold/heat shock proteins (CSPs/HSPs) and antifreeze proteins (AFPs)], biosynthesis of compatible solutes, structural adjustment of enzymes, and membrane fluidity, are survival mechanisms that allow microbial cells to resist extreme temperatures [87,88]. For survival under thermal stress, the MoT species of the genera *Kocuria*, *Methylobacterium*, *Bacillus*, *Pseudomonas*, *Acinetobacter*, *Micrococcus*, *Moraxella*, and *Sphingomonas* have developed at least one of those adaptation mechanisms [88,89,90,91].

**Desiccation resistance:** Water is crucial to basically all biological processes, such as cell structure stability, protein folding, and enzyme-substrate interactions, and living organisms apply several protective mechanisms to mitigate the damage caused by water loss, such as biofilm formation, sporogenesis, compatible solute biosynthesis, and accumulation [92,93]. In addition to those well-known mechanisms, microorganisms use other strategies to survive the desiccation, including the synthesis of stress-regulated proteins, deactivation of free oxygen radicals, prevention of protein glycation (the anti-glycation defense), and a series of strategies based on intrinsically disordered proteins (IDPs) [92].

**pH tolerance:** There are several common survival strategies to tolerate pH variations, including biofilm formation and extracellular polymeric substance (EPS) synthesis [94]. In addition, several particular strategies—including ferric iron respiration, synthesis of the impermeable membrane to protons, and the presence of squalenes, tetrahydrosqualenes, and other polyisoprenes in the membrane—are applied by acidophilic and alkaliphilic microorganisms in order to maintain the intercellular pH near neutral pH [95,96].

**Osmotic pressure:** Microorganisms use general and/or particular osmoadaptive mechanisms to balance the internal and external osmotic pressures [97]. The osmolytes’ synthesis—including proline, glycine betaine, carnitine, proline betaine, dimethylsulfoniopropionate, ectoine/hydroxyectoine, trehalose glucosylglycerol—and/or uptake of higher amounts of ions are two common osmoregulation mechanisms are harboring by microbial cells to survive the osmotic shocks [97,98]. Moreover, high osmotic pressure can emerge due to desiccation, freezing temperature, and high salt concentration [97]. Several specified osmoadaptive mechanisms have been observed in microbial cells that inhabit the extreme niches, including structural modification of biopolymers, proteoliposome systems, and the presence of a high number of osmosensors (osmosensing transporters) in the membrane [98,99].

## 4. Biotechnological Potential of the Microbiome of Things

As defined before, the MoT are those microbial communities that naturally inhabit anthropized artificial devices or machines with at least one feature that acts as a selective pressure for living organisms. As a result, these devices have a differentiated and unique microbiome that, through the adaptation mechanisms developed by the MoT, constitute potential sources of several biotechnologically relevant biomolecules, such as carotenoids, antifreeze proteins, biosurfactants, and hydrolysis enzymes.

**Carotenoid production:** As it was hinted, some microorganisms from artificial environments are carotenoid producers. Among them all, we can highlight from the photovoltaic panels, the *Arthrobacter*, *Deinococcus*, *Kocuria*, *Sphingomonas* and *Hymenobacter* strains and from the heating and AC systems, the *Methylobacterium*, *Sphingomonas* and *Moraxellaceae* species. Carotenoids are considered health-promoting, therapeutic molecules in the pharmaceutical industry due to their antioxidant activity, as well as good colorants in the food industry [81].

**Antifreeze and cold shock proteins (AFPs & CSPs):** Antifreeze proteins (AFPs) are synthesized by different species to avoid the formation of ice crystals in temperatures below the freezing point of water. These proteins have a variety of applications in the food industry, agriculture, cryo-surgery, and cryopreservation of cells and tissues [100,101,102]. Moreover, the presence of *Methylobacterium* sp. from Antarctica revealed that the genome of this bacteria includes many genes encoding cold shock proteins (CSPs) [103]. *Arthrobacter* sp., *Sphingomonas* sp., and *Flavobacterium* sp., members of the MoT we described here, are also AFP producers [100,104].

**Biosurfactants:** Biosurfactants have several advantages over chemical surfactants in terms of low toxicity, high biodegradability, activity and stability under extreme temperature, pH, and salinity [105]. Microorganisms, including genera *Rhodococcus*, *Pseudomonas*, *Pseudoalteromonas*, *Idiomarina*, *Sphingomonas*, *Bacillus*, *Marinomonas*, and *Halomonas*, adapted to cold natural habitats and are often especially suited for the production of biosurfactants. [106,107] However, several biosurfactant producers have also been isolated from hot natural environments: *Acinetobacter* sp., *Methanobacterium* sp., *Bacillus* sp., *Pseudomonas* sp., *Burkholderia* sp., *Enterobacter* sp., *Pantoea* sp., *Pseudoxanthomonas*, *Paenibacillus* sp., and *Anoxybacillus* sp. [108].

**Enzymes**: Although extremozymes from extreme natural environments have been deeply studied within the past decades, extreme artificial habitats have been less explored to date. To the best of our knowledge, only a few studies have focused on extremozymes in microorganisms isolated from man-manufactured devices [64] or studied saunas and dishwashers as sources of microorganisms with lipolytic activity within a broad range of temperatures [64].

## 5. As Conclusions: The Microbiome of Things

Artificial devices are not different from their natural counterparts, in the sense that their surfaces, internal liquids, or cavities can also be colonized by a wide range of microorganisms. Some of these devices are in constant interaction with humans (smartphones, remote controls) and thus share many of the taxa that characterize human gut and skin microbiomes [30,33]. The surface of home appliances, as part of the so-called “indoor environment”, is also strongly affected by the “house microbiome”, which, in turn, is derived from the microbial profile of the human and pet inhabitants of the space [35].

However, artificial objects, devices, and machines also contain an often-human-independent microbiome that is not necessarily similar to human microbiomes and on which we have focused in the present work. One of the most obvious case of an artificial object with a non-anthropogenic microbiome is solar photovoltaic panels [25,26], on which enteric or skin-associated bacteria are absent, and the main microbial key players are the same in cold and hot deserts or in polar microbial mats. The case of machines operated indoors is a bit different. The insides of home appliances such as coffee machines, dishwashers, or washing machines host microbial profiles that contain both human-associated and environmental microorganisms, the latter including genera such as *Enhydrobacter*, *Pseudomonas*, *Sphingomonas*, *Acinetobacter*, etc. (Table 1). When the selection pressure of a given machine, particularly in terms of temperature, clearly exceeds the usual (mesophilic) range, microbial populations tend to be totally environmental extremophilic ones, such as the *Thermus*-dominated water heating system microbiomes. Interestingly, many machine microbiomes combine extremophilic microorganisms (*Thermus* spp., *Deinococcus* spp., *Thermicanus* spp., etc.) and generalist ones (*Bacillus*, *Pseudomonas*, *Acinetobacter*, etc.). The presence of the former group suggests that colonization is, at least partially, air-mediated. For example, in the case of solar panels, bacteria-rich dust particles from deserts can be responsible for this colonization [109]. By contrast, the origin of the colonization of closed water systems by *Thermus* or *Acidovorax* may be less obvious.

Regardless of the origin of the microbial profile, it is clear that machines should not be seen as mere transporters of human or environmental microorganisms but as truly microbial niches. Like their natural counterparts (thermal springs, acidic environments, deserts, or sea waters), the combination of the biotic and abiotic factors in the devices acts as the main evolutionary driver shaping the Microbiome of Things.

## Figures and Tables

**Figure 1 microorganisms-11-01507-f001:**
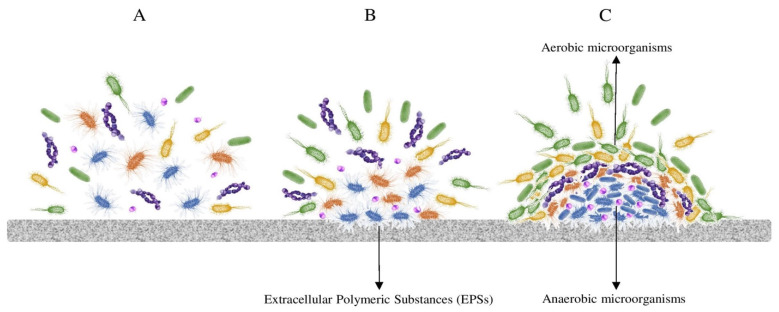
Biofilm formation is a mechanism that helps deal with polyextremophilic environments. (**A**) Microorganisms disperse by air, water, or human activities. (**B**) The extracellular polymeric substances (EPSs) producers attach to the surface, and (**C**) A multispecies biofilm can be formed, including various ecologically diverse layers by the cooperation of heterotrophs, chemotrophs, and/or phototrophs.

## Data Availability

No new data were created or analyzed in this study. Data sharing is not applicable to this article.

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
