# Peer review of "The Microbiome of Things: Appliances, Machines, and Devices Hosting Artificial Niche-Adapted Microbial Communities"

_microorganisms, 2023, doi:10.3390/microorganisms11061507_

Round 1

Reviewer 1 Report (Previous Reviewer 4)

The review article " The Microbiome of Things: Appliances, machines and devices hosting artificial niche-adapted microbial communities" is intriguing research on the association and adaptation of microbes on man-made materials.

The grammar and mistakes like extra dots need to recheck and corrected.

Picture-1: I have seen this picture before in an article. mention the source name if you copied the concept or the image.

-

Author Response

Point-to-point answer to Reviewer1
The review article " The Microbiome of Things: Appliances, machines and devices hosting artificial niche-adapted microbial communities" is intriguing research on the association and adaptation of microbes on man-made materials.

The grammar and mistakes like extra dots need to recheck and corrected.

The text was rechecked and corrected.

Picture-1: I have seen this picture before in an article. mention the source name if you copied the concept or the image.
The figure is original and generated by the authors. We summarised the results of previous research on biofilm formation in this figure. The similarity of this figure and others might be due to the fact that this is a classical way of representing biofilm formation on a surface, as described by previously published research in this field (References are available in the text).

Reviewer 2 Report (Previous Reviewer 3)

The review "The Microbiome of Things: Appliances, machines and devices hosting artificial niche-adapted microbial communities" is devoted to the systematization of information on the microbial community diversity of artificial niches and ways of their adaptation to environmental conditions. I hope that the comments below can help the authors to improve the review.

Comments:

The authors write that "The focus of the “Microbiome of Things” (MoT), the existence of which we propose here, is analogous to the “Internet of Things” (IoT) concept." What exactly is this analogousness? This fact should be clarified.

Additionally, I think the "Biotechnological potential of the Microbiome of Things" section in the context of this review is unnecessary. If it is important for the authors to mention the biotechnological potential of artificial niches, they can give some information about this in the introduction of the "Microbiomes on artificial devices" section and in the Conclusions.

Minor comments:

Since the absence of line numbers makes it difficult to provide specific comments, page numbers and section titles will be listed below to which authors should pay attention.

  1. page 2 subsection 2.1. - Unnecessary dot in the title of the subsection
  2. page 2 subsection 2.1. - In the sentence "Although not in close contact with humans once in place, previous works have made it clear that solar panels are not merely accumulating dust-borne microorganisms but instead, a selection takes place in situ, demonstrating that it is their own nature rather than the climate, what shapes the microbial community inhabiting them" -  references to these previous works (in bold type) should be added.
  3. page 3 subsection 2.2. - "The analysis of the bibliography clearly suggests that artificial devices with a selective pressure tend to have additional non-human associated, specifically adapted microbial taxa, that result in truly unique microbial niches (Table 1)." The phrase in bold is incorrect because Table 1 does not reflect the uniqueness of microbial niches, but provides information on the most common taxa on artificial surfaces.
  4. page 3 subsection "Coffee-machines" – "…by Vilanova et al. (2015) " should be changed to "[37]"
  5. page 3 subsection " Dishwasher" – In the sentence "Although fungi from the cluster Ascomycota followed by Basidiomycota …" -  "cluster Ascomycota" should be replaced by "phylum Ascomycota"
  6. page 5 subsection "Air conditioning systems (ACs) " - should be marked in bold.
  7. In Table 1, the bold type in the case of the "Moraxellaceae family" should be removed.

In addition, I think the authors should mention in the review that for some artificial niches there were no studies of microbial communities of fungi. Otherwise, readers might think that either the fungal community is not abundant or the most represented species cannot be indicated.

Minor editing of English language required

Author Response

Point-to-point answer to Reviewer2
The review "The Microbiome of Things: Appliances, machines and devices hosting artificial niche-adapted microbial communities" is devoted to the systematization of information on the microbial community diversity of artificial niches and ways of their adaptation to environmental conditions. I hope that the comments below can help the authors to improve the review.

Comments:
The authors write that "The focus of the “Microbiome of Things” (MoT), the existence of which we propose here, is analogous to the “Internet of Things” (IoT) concept." What exactly is this analogousness? This fact should be clarified.

We believe this is clearly clarified in lines 59 to 66.

Additionally, I think the "Biotechnological potential of the Microbiome of Things" section in the context of this review is unnecessary. If it is important for the authors to mention the biotechnological potential of artificial niches, they can give some information about this in the introduction of the "Microbiomes on artificial devices" section and in the Conclusions.

Thanks for the suggestion. However, we consider that the section “Biotechnological potential of the Microbiome of Things” should stay in the review, as it highlights for the reader the relevance of investigating the MoT from the perspective of potentially exploiting the unique features of the microorganisms adapted to the selective pressures of the devices they inhabit. We truly believe that the biotechnological potential of such specialised microbial communities is one of the biggest drivers of the microbial bioprospecting research field and what could in the future make a difference in society.

Minor comments:
Since the absence of line numbers makes it difficult to provide specific comments, page numbers and section titles will be listed below to which authors should pay attention.
Although we added the line numbers to the previous version, we realised that the numbers might have been removed by the editorial system. Sorry for the inconvenience!

page 2 subsection 2.1. - Unnecessary dot in the title of the subsectionpage

The extra dot was removed from the title.

 2 subsection 2.1. - In the sentence "Although not in close contact with humans once in place, previous works have made it clear that solar panels are not merely accumulating dust-borne microorganisms but instead, a selection takes place in situ, demonstrating that it is their own nature rather than the climate, what shapes the microbial community inhabiting them" - references to these previous works (in bold type) should be added.

Page 2, line 78: references were added.
page 3 subsection 2.2. - "The analysis of the bibliography clearly suggests that artificial devices with a selective pressure tend to have additional non-human associated, specifically adapted microbial taxa, that result in truly unique microbial niches (Table 1)." The phrase in bold is incorrect because Table 1 does not reflect the uniqueness of microbial niches, but provides information on the most common taxa on artificial surfaces.

To avoid any confusion on Page 3, lines 119-122, we added:

“The most common microbial taxa that colonize the following artificial devices are summarized in Table 1, although it should be taken into account that for some artificial niches there were no studies of the fungal communities. “
page 3 subsection "Coffee-machines" – "…by Vilanova et al. (2015) " should be changed to "[37]"

Page 3, line 129: “by Vilanova et al. (2015) " was changed to the relevant reference number "[36]".
page 3 subsection " Dishwasher" – In the sentence "Although fungi from the cluster Ascomycota followed by Basidiomycota …" - "cluster Ascomycota" should be replaced by "phylum Ascomycota"

Page 4, Line 152:  "cluster Ascomycota" was replaced by "phylum Ascomycota".
page 5 subsection "Air conditioning systems (ACs) " - should be marked in bold.

We revised that part in the text.
In Table 1, the bold type in the case of the "Moraxellaceae family" should be removed.
We revised that in Table 1.
In addition, I think the authors should mention in the review that for some artificial niches there were no studies of microbial communities of fungi. Otherwise, readers might think that either the fungal community is not abundant or the most represented species cannot be indicated.

This is an excellent suggestion. We have clarified this point in lines 120-122.

Minor editing of English language required.

We have reviewed the manuscript, and we hope the minor errors have been fixed.

Reviewer 3 Report (Previous Reviewer 2)

This resubmitted manuscript is a revised version of the previously submitted paper with the ID microorganisms-2263737. I am pleased to report that the authors have satisfactorily addressed all the concerns that I raised earlier. After thoroughly reviewing the revised manuscript, I have no further major comments regarding this submission.

However, there are a few minor corrections that should be made. In Table 1, "Solar panels" should be consistent with the name "Photovoltaic panels" used in the main body. Additionally, "Air conditioning systems (ACs)" should be in bold text, like other similar terms. 

Author Response

Point-to-point answer to Reviewer3
This resubmitted manuscript is a revised version of the previously submitted paper with the ID microorganisms-2263737. I am pleased to report that the authors have satisfactorily addressed all the concerns that I raised earlier. After thoroughly reviewing the revised manuscript, I have no further major comments regarding this submission.

Thank you!

However, there are a few minor corrections that should be made. In Table 1, "Solar panels" should be consistent with the name "Photovoltaic panels" used in the main body.

In Table 1 "Solar panels" was replaced with "Photovoltaic panels".

 Additionally, "Air conditioning systems (ACs)" should be in bold text, like other similar terms.
We revised that part in the text.

Reviewer 4 Report (Previous Reviewer 1)

I enjoy the authors' responses so that I decide this revised manuscript can be accepted.

Author Response

Point-to-point answer to Reviewer4
I enjoy the authors' responses so that I decide this revised manuscript can be accepted.

Thank you!

Round 2

Reviewer 2 Report (Previous Reviewer 3)

I would like to thank the authors for their adjustments made to the manuscript.

The manuscript may be accepted in its present form.

This manuscript is a resubmission of an earlier submission. The following is a list of the peer review reports and author responses from that submission.

Round 1

Reviewer 1 Report

This manuscript described many interesting concerns about artifical niche-adapted microbial communities and posed a novel topic of Microbiome of Things, however, there are still many flaws  to be corrected. 

General comments are listed as follows:

1. The authors should concern the risk of those adaptive microbes on appliances, machines and devices, especially for antibiotic-resistant bacteria.

2. The biotechnological potential of microbiome of things should be deeply discussed, such as antibiotic drug production, probiotics, and so on.

3. The author should dig deeply the meanings of the concept about microbiome of things.

Reviewer 2 Report

In recent years, there has been a growing awareness of the interconnectedness of microbiomes within ecosystems and their impact on human and environmental health. Despite this, research on the microbiomes of man-made objects and spaces has been relatively neglected. This review aims to address this gap by summarizing recent progress in the study of the "microbiome of things" (MoT), a concept proposed by the authors. While the study of microbiome of man-made objects and space is not entirely new, it has received little attention until now. By studying the microbial communities that exist on and within man-made objects, we can gain a better understanding of their role in shaping human health and the environment.

Overall, the review is comprehensive. One limitation of the review is that it tends to repeat existing research without providing a clear synthesis of the findings. For example, while the authors describe the microbial composition of various man-made objects, they do not adequately compare, explain, or summarize the differences between them. A more comprehensive and systematic overview of the microbial communities inhabiting different man-made objects would help readers understand MoT more comprehensively.

The authors emphasize that MoT should not be seen as mere transporters of human or environmental microorganisms, but as truly microbial niches. However, this point is not clearly stated in the text. The relationship and differences between MoT and other microbiomes should be more explicitly explained. The authors could do more to distinguish MoT from other microbiomes, such as the human microbiome or the built environment microbiome, and to explore the unique characteristics of MoT.

Regarding the mechanisms that allow microorganisms to thrive on man-made surfaces, the authors provide a general overview of microbial adaptation, including the formation of biofilms, but they do not delve into the specific adaptations that microorganisms make to different artificial surfaces of man-made objects. For example, a more detailed exploration of the effects of different materials on microbial colonization would be informative.

The discussion of the biotechnological potential of MoT is unclear. Clarifying the significance of MoT research would make the review more compelling and relevant to readers.

The numbering of sections in the manuscript is unclear, and the absence of line numbers makes it difficult to provide specific comments. It would be helpful if the authors could provide clear section headings and line numbers to facilitate the review process.

Reviewer 3 Report

OVERALL COMMENTS

 After careful reading and judgment, I think this research is not sufficiently innovative. For example, there is the following article, which explores a similar topic in detail:

1.      Novak Babič, M., Gostinčar, C. & Gunde-Cimerman, N. Microorganisms populating the water-related indoor biome. Appl Microbiol Biotechnol 104, 6443–6462 (2020). https://doi.org/10.1007/s00253-020-10719-4

There are also several significant comments that make it impossible to recommend this review for publication in its current form:

1)      The text does not indicate the Aim of the review. Why did the authors try to systematize this information?

2)      The absence of the aim makes it impossible to assess whether the introduction to the review is correct or not.

3)      In the section "Microbiomes on artificial devices" the authors wrote that they described the microbiological profile of "things" that were in constant contact with people. And in this case it is not clear why in this section there is information about the microbiome of photovoltaic panels in the subsection "Microbial diversity of sun-exposed artificial outdoor surfaces". In my opinion, photovoltaic panels are not in contact with humans, but only with the environment. They might as well have been reviewing the microbiome of traffic lights.  In addition, it is likely that the authors of the article on photovoltaic panels to which the reviewers are referring are not related to the microbiome of the panels after all, but to the microbiome of dust from the panels.

4)      A similar remark can be attributed to the section "Water heating systems". Since the authors of the review still focus on devices in constant contact with humans, the presence of the information described in the section Water heating systems is not clear to the end. These surfaces are not in direct contact with the human and are rather related to the water supply and water purification systems. In the same section there is information about saunas, which it is unclear how they are connected with devices?

5)      In the subsection "The microbial diversity of artificial devices and appliances," the authors should add information about the microbiome of the refrigerator and/or freezer as a machine in constant contact with humans and having extreme conditions for bacterial life (Jeon, YS., Chun, J. & Kim, BS. Identification of Household Bacterial Community and Analysis of Species Shared with Human Microbiome. Curr Microbiol 67, 557-563 (2013). https://doi.org/10.1007/s00284-013-0401-y).

6)      Microorganisms: from general to specialized metabolisms - this section does not describe metabolism as such, probably it should be called something else. Also, this section should have been combined with the section "Biotechnological potential of the Microbiome of Things". On the other hand, in the presented context of the review it is not clear why the last section (Biotechnological potential of the Microbiome of Things) is needed at all.

7)      Remarks on Table 1.

In the column "Typical marker genera" leave only really marker bacteria genera, because, for example, bacteria of the genus Pseudomonas are ubiquitous microorganisms and it is strange to see the genus Pseudomonas as a marker.

In the text to the description of Table 1, the authors need to add information describing the properties of these marker bacteria, which due to their adaptation are able to live in certain conditions of household appliances.

8)      Table 2 has no meaning and is essentially an abbreviated and modified Table 1.

 Overall conclusion:

The review is a poorly structured work. A serious revision of the review is required for it to be accepted for publication. I recommend that authors carefully revise the manuscript and then resubmit it. The absence of aim does not allow a qualitative assessment of the presented review. As a suggestion, the authors can revise their work by, for example, renaming it and conducting a thorough literature analysis on the topic such as: "The microbiome of the house: microbial communities of different rooms and their impact on human life and health". Then it could include some of the information presented not only on the microbiome of the interior surfaces of household machines, but also on the microbiome of dust in the rooms, on surfaces, microbiome from animals, in clothes, etc. And the purpose of the review would be related to the house and how it can affect human health.

Reviewer 4 Report

The article " The Microbiome of Things: Appliances, machines and devices hosting artificial niche-adopted microbial communities" is an interesting concept and research on microbial communities and their adaptations on devices we use every day.

It is a well-studied and precise review article. A very small number of corrections are needed.

The grammatical mistakes should be corrected. In the microbiome on artificial devices section line 7, 'the' should be removed. recheck the whole paper and correct it.

Please add the line numbers. it's hard to mention in the comment.

section 2.1

Line 4 - the proteomic concept is unwanted in that place or explain the associated protein or enzyme.